# Paths for Improving Bevacizumab Available in 2018: The ADZT Regimen for Better Glioblastoma Treatment

**DOI:** 10.3390/medsci6040084

**Published:** 2018-09-29

**Authors:** Richard E. Kast

**Affiliations:** IIAIGC Study Center, 22 Church Street, Burlington, VT 05401, USA; richarderickast@gmail.com; Tel.: +1-802-557-7278

**Keywords:** ADZT regimen, apremilast, bevacizumab, cAMP, dapsone, glioblastoma, telmisartan, TNF-alpha, zonisamide

## Abstract

During glioblastoma treatment, the pharmaceutical monoclonal antibody to vascular endothelial growth factor A, bevacizumab, has improved the quality of life and delayed progression for several months, but has not (or only marginally) prolonged overall survival. In 2017, several dramatic research papers appeared that are crucial to our understanding of glioblastoma vis-a-vis the mode of action of bevacizumab. As a consequence of these papers, a new, potentially more effective treatment protocol can be built around bevacizumab. This is the ADZT regimen, where four old drugs are added to bevacizumab. These four drugs are apremilast, marketed to treat psoriasis, dapsone, marketed to treat Hansen’s disease, zonisamide, marketed to treat seizures, and telmisartan, marketed to treat hypertension. The ancillary attributes of each of these drugs have been shown to augment bevacizumab. This paper details the research data supporting this contention. Phase three testing of AZDT addition to bevacizumab is required to establish safety and effectiveness before general use.

## 1. Introduction

This paper presents the physiological basis of the ADZT regimen, a new proposed augmentation strategy to improve the effects of bevacizumab (Avastin™) during the treatment of primary glioblastoma. Bevacizumab is a monoclonal pharmaceutical antibody directed against vascular endothelial growth factor A (VEGF). Initially the Food and Drug Administration (FDA) and European Medicine Agency (EMA) approved to treat some forms of macular degeneration, and it is now also approved for, and commonly used during, glioblastoma treatment after resection, radiation and temozolomide. Initial clinical studies of glioblastoma showed that bevacizumab delayed progression from ~7 to ~10 months, but did not impact overall survival (~16 months) [1]. Others found similar results [2]. Newer bevacizumab regimens with 100 mg/m^2^/day cycles of temozolomide and newer studies of lower bevacizumab doses have indicated some survival benefits [3]. Glioblastomas have been an unusually treatment-refractory cancer, justifying our exploration of unproven but low risk regimens like ADZT.

Multiple clinical trials have attempted, yet failed, to augment bevacizumab in prolonging overall survival. For example, in 2017 alone, clinical studies adding vorinostat, a histone deacetylase inhibitor [4], lomustine [5] and onartuzumab [6], all failed to prolong survival.

Crucial papers appeared in 2017 on the physiology of bevacizumab, each giving new data, and each independently converging on potential improvements to the bevacizumab treatment. By using four older drugs that are available now (mid-2018) we might be able to exploit these new insights to improve bevacizumab’s effectiveness in treating glioblastomas. The 2017 papers have been coalesced and integrated to form the ADZT regimen outlined here.

The four drugs used in the ADZT regimen are apremilast—an anti-psoriasis drug, dapsone—an antibiotic, zonisamide—an anti-seizure drug, and telmisartan, marketed to treat hypertension. While three are cheap, generic, and widely available, the fourth drug, apremilast, remains proprietary and somewhat expensive. The ancillary attributes of each, and the physiology of their interactions with bevacizumab’s effects, are detailed below.

All of the ADZT drugs are currently FDA or EMA approved, but are not marketed specifically for use in the augmentation of bevacizumab. Controlled phase two and three studies in a formal research setting, on the step-wise addition of ADZT drugs to bevacizumab, must be completed to establish safety and potential effectiveness prior to general clinical use.

## 2. Apremilast

Introduced to clinical practice in 2004, apremilast is a 461 Da, selective phosphodiesterase (PDE) 4 inhibitor. There are over a dozen currently-recognized isoforms of PDE. The problem with some past studies of pan-PDE inhibitors like pentoxifylline was that some PDE inhibitors have substrates that result in opposite intracellular effects to other PDE inhibitors. Since PDE4’s predominant intracellular role is to catalyze the reaction of cyclic adenosine monophosphate (cAMP) to AMP, apremilast results in increased intracellular cAMP. The cAMP is synthesized by ATP conversion to cAMP, mediated by adenylate cyclase. Multiple pro-inflammatory cytokines are partially inversely controlled by intracellular cAMP levels [7,8]. As intracellular cAMP decreases, synthesis and release of tumor necrosis factor alpha (TNF-α), interleukin-2 (IL-2), IL-8, and interferon-gamma tend to increase [7,8].

In accordance with these theoretical considerations, IL-6, IL-8, monocyte chemoattractant protein 1 (MCP-1), and TNF-α were reduced in people being treated with 20 mg of apremilast twice daily for psoriasis or psoriatic arthritis [8,9]. Since these cytokines have also been shown to participate in glioblastoma growth facilitation, we might expect benefits from apremilast on this basis alone.

Apremilast is now being used to successfully treat psoriasis [9,10,11] and psoriatic arthritis [12], atopic dermatitis [13], lichen planus, Behçet disease [14], ankylosing spondylitis [15], discoid lupus [16], chronic cutaneous sarcoidosis [17], and other inflammatory dermatoses.

Apremilast is generally well tolerated, with mild nausea, diarrhea and headache being the most common side effects. Discontinuation due to side effects was 5% with placebo and 7% with apremilast [12]. Another PDE4 specific inhibitor, rolipram, was investigated in the 1980s as an antidepressant; however, development stopped due to excessive nausea [18]. Rolipram inhibited the growth of A172 and U87MG glioblastoma cell lines through a PDE4 mediated pathway [19].

In March 2017, Ramezani et al. published a crucial paper for our next step in improving glioblastoma treatment by improving the effectiveness of bevacizumab [20]. They showed that adding a PDE4 inhibitor, rolipram, to bevacizumab enhanced in vitro cytotoxicity and reduced free VEGF in the culture medium, compared to bevacizumab alone. This finding makes sense in the larger general context of pro-inflammatory cytokine release.

Understanding that VEGF action might also be inversely related to intracellular cAMP opens several exciting augmentation pathways by which we might make bevacizumab more effective in treating glioblastoma. Alternatively, diminished free VEGF after rolipram could be a secondary effect of the previously established reduction of TNF-alpha, IL-8 and other cytokines by PDE4 inhibition.

Therefore, PDE4 inhibitors have evidence of (a) augmentation of bevacizumab effects, and independent of that, (b) anti-glioma growth effects and (c) lower synthesis of inflammation-related cytokines secondary to increased intracellular cAMP.

Apremilast would be a low-risk addition to bevacizumab.

## 3. Bevacizumab

Introduced clinically in 2004, bevacizumab is commonly called an anti-angiogenic agent, but it should be more accurately termed what it simply and literally is—a monoclonal humanized antibody to soluble VEGF. Beyond direct vessel effects, bevacizumab strongly suppressed the glioblastoma cell expression of 130 kDa platelet endothelial cell adhesion molecules and slightly reduced proliferation but upregulated matrix metalloproteinase-2 production [20]. Further, bevacizumab is cytotoxic (in vitro at least) to VEGF, synthesizing glioblastoma cells by binding to the outer cell membrane bound VEGF [21].

When a glioblastoma progresses while on bevacizumab, survival is under half a year [22,23,24]. Performance status and quality of life usually improve with bevacizumab; however, overall survival does not. ADZT aims to address this discrepancy.

Distorted, flawed vessels are common in glioblastomas. The pruning of these pathologic vessels occurs during bevacizumab treatment with a consequential reduction of tumor-related brain tissue edema [24]. However, an interesting paradox occurs here—vessel density and vessel morphological and functional abnormality decrease under bevacizumab treatment, yet hypoxia seems to increase [25].

## 4. Dapsone

Introduced in the mid-1940s, dapsone is a 248 Da sulfone antibiotic still in wide use. In addition to antibacterial activity in treating Hansen’s disease and pulmonary tuberculosis, dapsone has anti-protozoal effects and is currently used in the treatment of Plasmodia infections. Unrelated to antibiotic activity, dapsone has found some utility in treating neutrophilic dermatoses like bullous pemphigoid, dermatitis herpetiformis and others, including the neutrophilic rash caused by epidermal growth factor receptor inhibiting drugs [26]. In a series of five papers, my colleagues and I have amply documented the rationale for using dapsone to deprive the tumors of neutrophil-delivered VEGF during the treatment of glioblastoma [27,28,29,30,31].

As predicted in 2015 and in 2016 [27,28], dapsone was shown to an ameliorate anti-epidermal growth factor receptor mediated rash in 2017 [32,33], a rash mediated by VEGF containing neutrophils drawn to rash areas by IL-8 during erlotinib or cetuximab treatment, but countered by dapsone. We therefore expect dapsone to augment bevacizumab by reducing neutrophil borne VEGF in glioblastomas.

Dapsone has some in vitro anti-glioma activity on its own [34].

## 5. Zonisamide

Introduced in 1993, zonisamide is a 212 Da anti-seizure drug with carbonic anhydrase (CA) inhibitory activity that also blocks voltage-sensitive Na^+^ channels and T-type Ca^++^ channels [35,36]. There are a dozen CA isoforms. In vitro, the carbonic anhydrase IX (CA IX) Ki of zonisamide is 5.1 nM [37]. Zonisamide, unique among anticonvulsants, also inhibits monoamine oxidase [38].

Carbonic anhydrase catalyzes the reversible hydration of carbon dioxide to bicarbonate and a proton (H_2_O + CO_2_ ↔ HCO_3_^−^ + H^+^). Of the many isoforms of CA active in cancer physiology, CA IX is particularly prominent, including in glioblastomas [39,40,41,42]. CA IX resides on the outer cell membrane’s exterior. The resulting bicarbonate ion is imported by various pumps such as the Na^+^/HCO_3_^−^ cotransporter, raising intracellular pH but lowering extracellular pH as the proton remains extracellular. This is one of the primary mechanisms generating cancer’s—and specifically glioblastoma’s—abnormal extracellular acidic milieu. 

Concordant with the above mechanism of cancer-related extracellular acidification, topiramate, an anti-seizure and CA IX inhibiting drug similar to zonisamide, increased intracellular glioblastoma pH [43].

An immunohistochemistry study of grades II, III and IV glioma biopsy tissue by Yoo et al. found that they had strong CA IX expression, 21%, 33% and 79%, respectively [41]. The degree of CA IX expression inversely correlated with survival in this and in other similar studies [41,44]. Higher CA IX expression facilitates more vigorous in vitro growth of glioblastoma cell lines and in a xenograft glioblastoma model [45]. In this xenograft model inhibiting CA IX converted non-response to bevacizumab to growth inhibition by bevacizumab [45]. 

Remarkably, CA IX was absent in normal brain but expressed in 100% of human glioblastomas [39]. In that study Proescholdt et al. found that high CA IX expression had 15-months median survival compared to 34 months in patients with low tumor CA IX expression [39]. In a similar study Cetin et al. found median survival of 18 months in low CA IX expressing glioblastomas compared to 9 months with high expression [46]. These alone would seem to justify a clinical trial of already-marketed and well-tested CA IX inhibitors like acetazolamide, topiramate, and zonisamide.

In light of that inverse correlation, the conversion of a high CA IX expression glioblastoma to a poor CA IX functioning tumor by zonisamide may well prolong survival.

Both an experimental CA IX inhibitor and temozolomide individually inhibited the growth of human glioblastomas that were xenografted in nude mice. The effect was synergistic when used together [47]. The FDA approved pan-CA inhibitor acetazolamide augmented temozolomide cytotoxicity to glioma cells in vitro [48].

Acetazolamide reduces the production of cerebrospinal fluid (CSF) and is used clinically for this purpose [49], thus forming another potential benefit for the use of CA IX inhibitors like zonisamide during glioblastoma treatment, in addition to the potential augmentation of bevacizumab.

Dexamethasone use tends to worsen prognosis in glioblastomas [50] but must be used to decrease elevated CSF pressure during the course of glioblastomas. Since we expect dapsone and zonisamide will lower the need for steroids, we might see an overall survival increase on that account as well.

Acetazolamide is a sulfonamide pan-CA inhibitor that has had continuous clinical use since the 1950s with demonstrable preclinical anti-glioma activity [51,52]. However, to date, there have been no clinical trials in human glioblastomas, other than to treat plateau waves [53], as far as I was able to determine. Acetazolamide is currently used clinically to treat mountain sickness, elevated intraocular pressure and pseudotumor cerebri syndrome [54,55]. Acetazolamide could be substituted for zonisamide in an ADZT-type regimen.

The use of CA IX inhibition with zonisamide (or acetazolamide) would be a realization of Koltai’s “repurposed drug combinations targeting this vulnerable side (i.e., decreased extracellular pH and need to export increased intracellular protons) of cancer development” [56].

Crucially for our intended use of zonisamide, the lower the CA IX activity is in a given tumor tissue, the more effective bevacizumab becomes [44,45,57,58,59,60,61].

## 6. Telmisartan

Telmisartan is an angiotensin receptor blocking drug (ARB) with several unique features that recommend its use in glioblastomas, particularly in combination with bevacizumab. ARBs, like angiotensin converting enzyme (ACE) inhibitors, are marketed for a variety of indications, prominently hypertension. Telmisartan is uniquely lipophilic, has a tighter affinity to the angiotensin 2 type 1 receptor, and happens to inhibit peroxisome proliferator-activated receptor-γ (PPAR-γ) as well [62,63]. All of these attributes would be useful during the treatment of glioblastomas, particularly in co-administration with bevacizumab.

In 2017, Levin et al. suggested adding an ARB or ACE inhibitor to bevacizumab based on their retrospective glioblastoma study showing the overall survival of ~25 months in those receiving low dose bevacizumab plus an ARB or ACE inhibitor, compared to ~14 months for those receiving only a low dose bevacizumab [64]. They notably found that those receiving <3.6 mg wk/kg bevacizumab did better than those getting ≥3.6 mg wk/kg. This inverse dose-response relationship is currently (as of 2018) unexplained and is a huge hint in our efforts to understand how bevacizumab works.

Additionally in 2017, Menter et al. found similar but slightly different results in non-squamous, non-small cell lung cancer treated with carboplatin and paclitaxel with or without bevacizumab. Bevacizumab prolonged survival, as did an ACE inhibitor or ARB, but increased survival by the addition of an ACE inhibitor/ARB to bevacizumab did not reach statistical significance for additive effect [65].

A potential added benefit of adding telmisartan is that it is also a PPAR-γ agonist and PPAR-γ agonism has significant glioblastoma growth inhibiting effects [66].

In metastatic colon cancer, those receiving bevacizumab with an ARB compared to those receiving bevacizumab only had longer progression-free survival, eight versus six months, and longer overall survival, 26 versus 16 months [67].

## 7. Limitations and Conclusions

This paper outlined past human, in vitro, and animal studies showing evidence of the physiologic pathways by which the glioblastoma inhibiting effects of bevacizumab might be enhanced. The four drugs of ADZT engage these enhancing pathways. The ADZT drugs—apremilast, dapsone, zonisamide, and telmisartan—are low risk drugs when used individually, are inexpensive, and are generally well known and available to physicians worldwide.

Below is a list of the potential or expected benefits of the ADZT regimen during the course of glioblastomas:Lower intracranial pressure.Steroid sparing.Augment bevacizumab effect.Provide synergy with temozolomide.Provide inherent anti-glioma effects.Individually have low side effects, low risk.

There are several factors that may limit ADZT effectiveness. Hence our need for empirical, clinical evidence of the benefits established in a phase three trial. A main potential limitation of the ADZT regimen is the uncertain degree to which apremilast penetrates the blood-brain barrier. Dapsone, zonisamide and telmisartan are known to penetrate well.

The cancer research literature is replete with data on inflammation as both cancer causing and cancer sustaining, as well as data on inflammation as a cancer cell destruction element and a defense against cancer. A one-year phase three trial of apremilast in plaque psoriasis found no evidence of immune system dysfunction [68]. Apremilast treatment of psoriatic arthritis reduced the plasma inflammation markers of IL-8, TNF-α, IL-6, macrophage inflammatory protein 1β (MIP-1β), monocyte chemoattractant protein 1, (MCP-1), and ferritin [69]. Notably, all of these markers are precisely the inflammatory markers that are documented as being among the core pathophysiologic drivers of glioblastoma growth [70]. This, plus the demonstrated enhancement of bevacizumab’s anti-VEGF function and the synergistic decrease in glioblastoma cell viability by adding rolipram to bevacizumab [20], and the inherent documented anti-glioblastoma effects of rolipram alone [71], warrant risks of a phase three trial of adding apremilast to bevacizumab during primary glioblastoma treatment as part of the ADZT regimen. Unfortunately, there is no adequate murine model to test ADZT.

The ADZT regimen follows other efforts to improve the anti-glioblastoma effects of bevacizumab. Adding the chemokine receptor CXCR4 inhibitor plerixafor for example, did not improve survival over bevacizumab but did provide some clues as to the resistance or circumvention pathways around the anti-VEGF effects of bevacizumab-plerixafor plus bevacizumab increased CXCL12 (SDF-1) [72]. As seen in many cancer chemotherapies, the exposure of glioblastomas to bevacizumab engages tumor growth enhancing compensatory pathways like CXC12 in addition to intended growth inhibition. The ADZT regimen was designed to enhance bevacizumab mediated growth inhibition by blocking several of these circumvention pathways.

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
