# Peer review of "Paths for Improving Bevacizumab Available in 2018: The ADZT Regimen for Better Glioblastoma Treatment"

_medsci, 2018, doi:10.3390/medsci6040084_

Round 1

Reviewer 1 Report

This very interesting concept paper introduces the combination of bevacizumab with the four already established drugs apremilast, dapsone, zonisamide, telmisartan.

The failure of bevacizumab to prolong overall survival in several phase III studies was disappointing to many clinicians. The combination of bevacizumab with other drugs may allow for an escape from the difficult therapeutic situation in glioblastoma in the relapse situation. Therefore this manuscript will find many interested readers. I strongly support publication in "medical sciences". However, I fear that patients may be treated outside of clinical studies with experimental therapy regimes as the ADZT regime. Therefore I strongly request that 3 issues must be adressed:

Please add a sentence that such combinations must be tested in clinical studies prior to treatment of a large number of patients.

Please state that clinical studies phases I, II and III must be carried out with the ADZT regime prior to a more broad application.

Please discuss if there is preclinical in vivo data (a glioblastoma mouse model) available for the ADZT regime. If no preclinical data from mouse experiments is available, please clearly say that this data does not exist.

Author Response

Thank you for your kind words and helpful suggestions. I have addressed all your suggestions and areas of concern. All changes are highlighted in green.

Reviewer 2 Report

This is an interesting concept paper describing the use of 4 old drugs in combination with bevacizumab to treat a highly refractory type of cancer, namely, glioblastoma. The authors present convincing evidence that an AZDT regimen can potentiate the effects of bevacizumab resulting in improved inhibition of GBM growth. Given the dearth of current treatment options for GBM patients, this concept paper will likely be of interest to a broad audience of those interested in treatment of this deadly disease. I do however have some comments and suggestions that may improve the manuscript.

Please clarify whether the ADZT/bevacizumab regimen is intended for use in primary or recurrent GBM and whether it's been demonstrated to be more effective in one or the other.

Please address whether each of the drugs are able to cross the blood-brain barrier, or better yet, the blood-brain-tumor barrier. A cursory search in Pubmed revealed that apremilast has limited penetration of the BBB and that dapsone actually restores its' integrity.  Would this not limit the activity of telmisartan, which is know to readily penetrate the BBB? Also, apremilsat has been shown to negatively modulate immunity.

Please discuss how this might impact potential immunotherapy treatments currently targeting glioblastoma.

Author Response

Thank you for your kind words. I have addressed your concerns in the green highlighted areas. Your thought regarding dapsone potentially tightening the BBB, we agree. Dapsone is being investigated in glioblastoma precisely on this account. By tightening the loose, faulty vasculature of glioblastoma, dapsone may slightly reduce the characteristic edema of glioblastoma. We know that zonisamide and telmisartan readily enter the brain through healthy, normal brain capillaries. The poor penetration of apremilast is indeed a potential weakness and this comment has been added to the text.

Round 2

Reviewer 1 Report

This manuscript is now an excellent addition to the scientific literature which will find many readers. I just found one mistake:

Apremilast treatment of psoriatic arthritis reduced the reduced

Reviewer 2 Report

In the last highlighted paragraph, line 4, please change "Apremilast treatment of psoriatic arthritis reduced the reduced plasma inflammation markers..." to Apremilast treatment of psoriatic arthritis reduced the plasma inflammation markers....